# *One* or *Two Roots*? Yi Zhi and the Dilemma of Practical Reason

Xiaodong Xie

Philosophy Department, Xiamen University, Xiamen 361005, China; xxdong@xmu.edu.cn

**Abstract:** Mohism has two versions of ethics, attributed to Mozi and Yi Zhi 夷之, respectively. Mozi introduced an ethics usually described as utilitarian, emphasizing universal love as the basis of impartiality. However, the problem with this emphasis is that it leads to neglecting the development of rational self-interest. Accordingly, Yi Zhi's remarks are a clarification or modification of Mozi's thoughts. First, Yi Zhi alluded to the concept of undifferentiated love to explain universal love as the basis of impartiality. Second, as he understood the concept of undifferentiated love in relation to the idea that "bestowing love begins with one's parents", Yi Zhi incorporated rational self-interest. Moreover, Mencius criticized Yi Zhi and disparaged his remarks as *two roots* (二本 *er ben*), contrasting it to Confucian ethics, which he said was *one root*. This division between *one root* (一本 *yi ben*) and *two roots* has garnered significant attention. On the one hand, Zhu Xi believed that the essence of two roots is undifferentiated love, wherein he concluded that it is applicable to both Mozi and Yi Zhi. On the other hand, most later scholars interpreted two roots from an ethical perspective, arguing that Yi Zhi faced the dilemma of two conflicting moral theories. Considering the basic principles of moral philosophy, the ethics of Mozi and Mencius are one root, and only that of Yi Zhi is two roots. This article shows that Yi Zhi and Henry Sidgwick, the founder of classical utilitarianism, face the same dilemma of practical reason: the conflict between utilitarianism and the self-interest of egoism.

**Keywords:** Yi Zhi; Mozi; Mencius; utilitarianism; impartiality; *one root*; *two roots*; practical reason

## 1. Introduction

The terms *one root* (一本 *yi ben*) and *two roots* (二本 *er ben*) originate in the *Book of Mencius* (孟子 *Mengzi* 3A5).[1] The text reads:

[Yi Zhi], a Mohist, sought to meet Mencius through the good offices of [Xu Bi]. "I wish to see him too", said Mencius, "but at the moment I am not well. When I get better, I shall go to see him. There is no need for him to come here".

Another day, he sought to see Mencius again. Mencius said, "Now I can see him. If one does not put others right, one cannot hold the Way up for everyone to see. I shall put him right. I have heard that [Yi Zhi] is a Mohist. In funerals, the Mohists follow the way of frugality. Since [Yi Zhi] wishes to convert the Empire to frugality, it must be because he thinks it the only honorable way. But then [Yi Zhi] gave his parents lavish burials. In so doing, he treated his parents in a manner he did not esteem".

[Xu Bi] reported this to [Yi Zhi]. "The Confucians", said [Yi Zhi], "praised the ancient rulers for acting 'as if they were tending a newborn babe'. What does this saying mean? In my opinion, it means that there should be no gradations in love, though the practice of it begins with one's parents".

[Xu Bi] reported this to Mencius. "Does [Yi Zhi] really believe", said Mencius, "that a man loves his brother's son no more than his neighbor's newborn babe? He is singling out a special feature in a certain case: when the newborn babe creeps towards a well it is not its fault. Moreover, when Heaven produces things, it gives them a single basis [*yi ben*], yet [Yi Zhi] tries to give them a dual one [*er ben*]. This accounts for his belief.

"Presumably there must have been cases in ancient times of people not burying their parents. When the parents died, they were thrown in the gullies. Then one day the sons passed the place and there lay the bodies, eaten by foxes and sucked by flies. A sweat broke out on their brows, and they could not bear to look. The sweating was not put on others to see. It was an outward expression of their innermost heart. They went home for baskets and spades. If it was truly right for them to bury the remains of their parents, then it must also be right for all dutiful sons and benevolent men to do likewise".

[Xu Bi] repeated this to [Yi Zhi] who looked lost for quite a while and replied, "I have taken this point".

This well-known passage, which has gained considerable attention, recounts the encounter between the Mohist Yi Zhi 夷之 and Mencius. Most scholars think that Confucian ethics as represented by Mencius is *one root*, and Mohism or Yi Zhi's ethics is *two roots*. In contrast, in this article, we explore and reinterpret *two roots*—or, in general, the discussion that transpired in the encounter between Mencius and Yi Zhi—from the perspective of the dilemma of practical reason. We argue that Yi Zhi's *two roots* problem falls into a famous dilemma of moral philosophy: the dualism of practical reason.

## 2. Two Versions of Mohist Ethics: Mozi and Yi Zhi

In the above passage, Mencius criticizes Yi Zhi for violating the Mohist doctrine. Mozi (墨翟 *Mo Di*), the founder of Mohism, introduced the doctrine of frugality in funerals, which Yi Zhi, as a Mohist, must follow. However, he disobeys it and instead buries his parents lavishly. In response to this criticism, Yi Zhi states "there should be no gradations in love, though the practice of it begins with one's parents" (爱无差等, 施由亲始 *aiwuchadeng, shiyouqinshi*).[2] Moreover, this person, Yi Zhi, is not mentioned outside of the *Book of Mencius*. Hence, Liang Qichao 梁启超 (Cai 2008) asserts that it would be difficult to trace this person's lineage. Before analyzing why Yi Zhi understands the concept of undifferentiated love in relation to the idea that "bestowing love begins with one's parents", it is necessary to examine Mozi's ethics, and various challenges to it.

### 2.1. The Nature of Mozi's Ethics: Utilitarianism

Since the time of Liang Qichao, Hu Shi 胡适, and Feng Youlan 冯友兰, most Mainland Chinese scholars have referred to Mozi's ethics as utilitarian. Since then, overseas scholars (including those in Hong Kong and Taiwan scholars), such as Benjamin I. Schwartz, A.C. Graham, David Nivison, Lao Sze-kwang 劳思光[3] and Wei Zhengtong 韦政通 have also endorsed this position. Indeed, Mozi's ethics is in line with the basic ideas of classical utilitarianism.[4] It is generally accepted that the core of Mozi's teachings is "universal mutual love and exchange of mutual benefit" (兼相爱、交相利 *jianxiangai jiaoxiangli*; *Mozi* 26.4).[5] Additionally, various scholars have argued that Mozi's fundamental principle is "universal love" (兼爱 *jianai*).[6] Inspired by this principle, the Mohists proposed the ten core theses, also called the ten doctrines. These are: elevating the worthy (尚贤 *shangxian*), exalting unity (尚同 *shangtong*), impartial concern (兼爱 *jian ai*), opposing military aggression (非攻 *feigong*), frugality in expenditures (节用 *jieyong*), frugality in funerals (节葬 *jiezang*), Heaven's will (天志 *Tianzhi*), elucidating the spirits (明鬼 *minggui*), opposing music (非乐 *feiyue*) and opposing fatalism (非命 *feiming*).[7] As mentioned, Yi Zhi's lavish burial of his parents contradicts one of the doctrines of Mohism—frugality in funerals. This is the first point in Mencius' criticism to which Yi Zhi responds. To understand the strength of Yi Zhi's response, it is imperative to explain the fundamentals of two Mohist doctrines: utilitarianism and universal love. The central claim of utilitarianism is maximizing consequences, in which, whether an action is right or wrong depends on the maximization of utilities; it is the principle of the "greatest happiness for the greatest number of people" (GHP).[8] In Mohism, this GHP may be stated as: "generating the wellbeing of all people under heaven" and "eradicating the suffering of all people

under heaven" (see *Mozi* 15.1). For Mozi, 天下 *tianxia* or "all people under heaven" is the most encompassing subject. Thus, Mozi thinks it is important to consider the fate of this communal reality.

To attain the state in which "that all people under heaven may experience great benefits", Mozi recommends universal love. In short, universal love is the means to achieve the general good of humankind. In the language of moral philosophy, the essence of universal love is impartiality (不偏不倚 *bupian buyi*). Impartiality is an important component of utilitarianism, and an innovative concept (see Xu 2011, p. 12). As Mozi emphasized impartiality, he simultaneously opposed otherness (别 *bie*). Otherness appears to constitute the Confucian idea of graded love (爱有差等 *aiyouchadeng*), which presupposes that love for parents and other family members must exceed love for others. Since impartiality is regarded as a universal idea, many scholars believe that Mohism provides a vision and principles superior to those of Confucianism (Roetz 1993). Moreover, universal love requires that people be neutral and impartial because it is only in this way whereby GHP can be attained. Conversely, since graded love is unequal, it undermines the GHP. It is also important to note that Mozi's universal love is a modification of the Confucian ideas of graded love and benevolence (仁爱 *renai*; see Zhu 1983, p. 262; Yang 2017). Because universal love is altruistic, Mohism appears more demanding than Confucianism. Mozi's original version of universal love is equal and impartial, which is reflected in Yi Zhi's response. Although Yi Zhi includes "bestowing love begins with one's parents" in his response to Mencius, it is important to point out that Yi Zhi's understanding of undifferentiated love implies Mozi's universal love.

*2.2. The Fundamental Challenge to Mozi's Ethics: Moral Demand Is Too High*

There are two theoretical criticisms of utilitarianism: the requirement of maximizing consequences, and the point of view of impartiality. In sum, these two criticisms suggest that utilitarianism's moral demand is too high. The first criticism involves the question of rationality, which is implied, but not emphasized, in early Chinese philosophy. Therefore, this article focuses on the second criticism. In utilitarianism, impartiality was initially introduced in the concept of the "impartial spectator" proposed by Adam Smith (2002). The crux of this concept is seeking a purely rational vision to calculate utility, and thus arrive at rational choices. Like classical utilitarianism, Mozi faced a similar problem, evident in his encounter with his contemporary, Wu Mazi 巫马子. Wu Mazi doubts his own ability to practice universal love. In the text, he asks: "[universal love] may be good [i.e., benevolent and righteous, 仁 *ren* and 义 *yi*], but how can it be put to use?" (*Mozi* 16.5). Here, Wu Mazi suggests that universal love is too demanding to be implemented. Meanwhile, Wu Mazi's doubt is consistent with Zhuangzi's 庄子 later observations about Mozi. According to Zhuangzi (33.2):[9]

> [Mozi's view] just brings sorrow and worry to the people. I fear this can never be used as the Course of the Sage. The people of the world cannot endure such a thorough rejection of what is in their own hearts. Although Mozi himself may have been up to the task, what use is that for the rest of the world?

In general, Zhuangzi thinks that the ten doctrines of Mozi are too demanding and are difficult to universalize. Accordingly, Zhuangzi also thinks that Mozi's universal love cannot be universalized. Although Mozi can do it, it cannot be forced upon others; otherwise, it would be "a thorough rejection of what is in their own hearts". Mozi's claim can be held as an individual aim, but it cannot be imposed on others as their aim. In this regard, Zhuangzi opposed Mozi's requirement to "do unto others as you would have them do unto you". Since it is considered unacceptable, Mozi's views cannot be implemented. As Zhuangzi (33.2) states, referring to Mozi's views, "I'm afraid that to instruct people thus shows no real love for them. And to put it into practice personally certainly shows no real love for oneself!" This discussion suggests that some people find Mozi's moral demands too high, and therefore unacceptable. However, the above challenges only highlight the problems with Mozi's ethics; they do not recommend alternatives.

The significance of Yang Zhu's 杨朱 notion of "each for himself" (为我 *wei wo*) is better understood in this context. Regarding the question of chronological sequence, in this article, we adopt the view that there was a Yang Zhu stage in Daoism that preceded Zhuangzi (see Fung 1948). Moreover, the *Huainanzi* 淮南子 presents the causal relationship between the thoughts of Mozi, Yang Zhu, and Mencius.[10] The text reads (*Huainanzi* 13.9):

> Universal love, honoring the worthy, esteeming ghosts, opposing fatalism: These were established by Mozi, but Yangzi [or Yang Zhu] opposed them. Keeping your nature intact, protecting your authenticity, not allowing things to entangle your form: These were established by Yangzi, but Mencius opposed them.

This text indicates that Yang Zhu's notion of "each for himself" is a response or an alternative to Mozi's universal love. What is ironic is that universal love loses the self, and without the self, it is impossible to love everyone. Perhaps, it is apt to say that Yang Zhu's objection springs from the perspective of individualism. This objection is the opposite of Zhu Xi's interpretation. In Zhu Xi's view, based on the *Mengzi*, Yang Zhu should precede Mozi. Here, Zhu Xi proposes that the principal aim of Mozi was to judge egoists such as Yang Zhu. As Zhu Xi 朱熹 (Li 1986, p. 1320) describes, "Mozi saw that people in this world are selfish, and they do not care for others, hence he proposes that all people under heaven should love each other". With these two different accounts, we accept the viewpoint of *Huainanzi*. Hence, Mozi preceded Yang Zhu (Sun 2001; Fung 1948). In this case, Yang Zhu can be considered Mozi's first challenger, i.e., egoism challenging altruism. From the perspective of moral philosophy, the essence of Yang Zhu's challenge is to accommodate an independent, autonomous self.

Mencius also challenged Mozi's universal love. In the *Mengzi*, apart from the quotation in this article's introduction, there are two other passages that also refer to Mohism. One of the passages states "if scraping himself [Mozi] bare from head to heels would benefit the whole world, he would do it" (*Mengzi* 7A26). This is to say that the Mohists are a group of people who are zealous for the good of the world. In the same vein, the *Huainanzi* also describes Mohist ethics. It states: "those who served Mozi numbered one hundred and eighty. He could send them all to walk through fire and tread on blades, face death, and not turn their heels [to flee]" (*Huainanzi* 20.22). In other words, Mohists can risk their lives. These passages seem to praise Mohists as moral saints. However, Mencius criticizes Mohists' universal love. He states (*Mengzi* 3B9):

> Since then, a sage King has not arisen; the various lords are dissipated; pundits engage in contrary wrangling; the doctrines of Yang Zhu and Mozi fill the world. If a doctrine does not lean toward Yang Zhu, then it leans toward Mozi. Yang Zhu is 'for oneself.' This is to not have a ruler. Mozi is 'impartial caring [or universal love].' This is to not have a father. To not have a father and to not have a ruler is to be an animal.

This is the criticism of Mohism's principle of universal love, which leads to extreme altruism, thus refusing priority to relatives. Later, Zhu Xi agrees with Mencius, Zhu Xi states (Zhu 1983, p. 272):

> Yang Zhu only knows how to love oneself, but he does not know that the self must practice righteousness, therefore he does not have a ruler; Mozi practices undifferentiated love and he regards his relatives as the same as everyone, therefore he does not have a father. Without a father or without a king, the way of being human becomes extinct, humans are like beasts.

One of the main points of Zhu Xi's criticism is that Mozi's undifferentiated love ignores the special moral relationship between relatives, and thus is a beast. Mencius and Zhu Xi accused Mozi of "fatherlessness", and some have questioned this. The text reads (Li 1986, p. 1320):

> The question: Regarding Mozi's universal love, why does it mean not to have a father? The response: A person has only one (set of) parents, and no one has seven hands or eight feet to love a lot. To support one's father decently is already

difficult. The reason why he supported his parents is he only wears coarse clothes and eats simple food, which he cannot bear. Desiring universal love, he cannot love his parents, and he cannot practice filial piety satisfactorily, therefore he will not have a father. Since Mozi values frugality and hates music, he reverses his chariot and went back when he heard morning songs linger around the lanes. No wonder he seeks no fame and fame is indifferent to him. It is imaginable how he treats his parents.

Considering human limitations, Zhu Xi pointed out that the principle of universal love will inevitably lead to a reduction of natural love for parents, and the love for parents will be even more tenuous. From the perspective of modern moral philosophy, the essence of Mencius and Zhu Xi's criticism lies in the fact that Mohism's universal love derives from an impersonal standpoint that emphasizes impartiality. Hence, Mohism cannot accommodate personal standpoints (see Nagel 1991).

### 2.3. "Bestowing Love begins with One's Parents": Revising Mozi's Universal Love

Based on the *Huainanzi* (13.9) mentioned above, the following sequence can be drawn: Mozi criticized Confucius, Yang Zhu criticized Mozi, and Mencius criticized Mozi. Considering the encounter between Mencius and Yi Zhi as stated in the *Mengzi*, this article provides a hypothesis: Yi Zhi responded to Yang Zhu's criticism by emphasizing self-interest (burying his relatives), while Mencius criticized Mozi and Yi Zhi, pointing out that Yi Zhi faced the dilemma of dualism. Of course, Yi Zhi's response to Yang Zhu requires modification of Mozi's insistence of impartiality to accommodate special relationships (see Dong 2015).[11] Since Confucianism prioritizes relationships such as kinship, Yi Zhi seems to lean towards Confucianism. In Mencius' view, Yi Zhi's lavish burial of his parents violates the doctrine of frugality in funerals, and this action undermines the essence of universal love. In short, Yi Zhi's actions mean that his love for his parents exceeds that of his love for others, thus violating universal love. In response to Mencius' criticism, Yi Zhi defends himself, stating "the Confucians [ . . . ]praised the ancient rulers for acting 'as if they were tending a newborn babe'. What does this saying mean? In my opinion, it means that there should be no gradations in love, though the practice of it begins with one parents" (*Mengzi* 3A5). One interpretation of this statement is the idea that "bestowing love begins with one's parents" suggests that Yi Zhi intends to weaken Mozi's claim of impartiality and emphasize the rational development of self-interest, thus modifying universal and undifferentiated love. Another possible interpretation is that Mencius accused Mozi of "fatherlessness", and Yi Zhi responded by burying his relatives. In this way, Yi Zhi augmented Mozi's utilitarianism as altruism by incorporating self-interest. However, in that case, Yi Zhi must be able to justify himself. As for Mencius' attack, he cites "affection for a child as if one's own" (若保赤子 *ruobao chizi*) as found in the *Book of Poetry* (诗经 *Shijing*) to defend himself. What is meant here is that the supreme ruler treats all of the ruled as if they were his children, and gives them equal love and care. Mencius' example supports the idea that all people have compassion for the children. In this way, Yi Zhi explains the universality of compassion by saying that it is the universal love preached by Mozi. Yi Zhi's approach is indeed clever. In addition, it also shows that he tries to find the foundation of impartiality in the human heart/mind (心 *xin*) and nature (性 *xing*), unlike Mozi who offers little discussion of heart/mind and nature.

Yi Zhi's rhetorical question, "what does this saying mean?" is a mockery of Mencius. Perhaps, it is apt to say that Yi Zhi's response justifies Mozi's universal love. In other words, Yi Zhi justifies universal love in relation to undifferentiated love. We argue that Yi Zhi is the first figure to concretize Mozi's principle of universal love by alluding to undifferentiated love. Such concretization clarifies and popularizes Mohism. More importantly, through this interpretation, the concepts of love emerging in Mohism and Confucianism, respectively, are distinguished. The Confucian virtue of benevolence (仁 *ren*) is an important innovation embodied in humanity (仁爱 *ren ai*). Mozi, who studied Confucianism, followed the idea of *ren* (see *Huainanzi* 21.4). However, in Mozi's view, *ren ai* is narrow; thus, he replaced it with

the principle of universal love. In other words, Mozi regarded Confucian *ren ai* as "classified love" or loving depending on who is being loved (别爱 *bie ai*), which Mozi criticized. Yi Zhi further concretized Mozi's criticism of Confucian *ren ai*. He equated graded love with Confucian *ren ai*, and then proposed undifferentiated love as synonymous with Mozi's universal love. Yi Zhi's observation is keen, and his contrasting of Confucianism and Mohism was recognized by Zhu Xi (Zhu 2001, p. 444).

### 3. The Dispute between *One Root* and *Two Roots*

Mencius was unconvinced by Yi Zhi's defense, for two reasons. Firstly, the ethical principles of Confucianism can be considered *one root*, while the ethical principles of Yi Zhi can be considered *two roots*. Secondly, the Confucian notion of filial piety is understandable from the point of view of moral psychology (Riegel 2015). Pertinent to this article's aim, we focus on the first reason.

#### 3.1. One or Two Roots?

Let us again recall the words of Mencius (3A5):

Does [Yi Zhi] really believe[ . . . ] that a man loves his brother's son no more than his neighbor's newborn babe? He is singling out a special feature in a certain case: when the newborn babe creeps towards a well it is not its fault. Moreover, when Heaven produces things, it gives them a single basis [*yi ben*], yet [Yi Zhi] tries to give them a dual one [*er ben*].

For Mencius, Yi Zhi was too naïve to think that a person could have the same love for his nephew and his neighbor's son. Mencius also refuted Yi Zhi's use of the Confucian ideal of "as if they were tending a newborn babe" (or this can also be understood as the "affection for a child as if (it is) one's own") to justify his love for both. While there is indeed universal compassion for an innocent child who is about to fall into a well, this cannot be used to prove universal love. In other words, the Confucian concept of compassion is thin; the Mohist concept of universal love is more substantial. Therefore, the former cannot be used to prove the latter. In general, the question of *one root* or *two roots*, as summarized by Mencius, underlies the fundamental difference between Confucianism and Mohism. The ensuing discussion focuses on this question.

#### 3.2. Zhu Xi's Understanding of Two Roots: Undifferentiated Love

The original meaning of the terms *one root* and *two roots* is unclear. Hence, some translators have dealt with them more literally (Lau 2003; Yang 1960). Commentators have expressed their views about the context and the whole text of *Mengzi*. The views of Zhu Qi 赵岐 and Zhu Xi are noteworthy. Zhao Qi is the earliest annotator of Mencius. He says, "Heaven gives birth to all things; each comes from one root. Now, Yi Zhi takes the parents of others as equal to his parents, which are then two roots, and therefore he desires to give them similar love". (see Jiao 1987). In Zhao Qi's view, all things are born in Heaven, with only *one* original *root*. As he values his parents in the same way he values others' parents, Yi Zhi juxtaposes *two roots*. The crux of Zhao Qi's explanation is that the love for one's parents and the love for the parents of others come from *two roots*. Since Yi Zhi interprets Mozi's universal love as undifferentiated love, Zhao Qi's description of Yi Zhi's *two roots* is similarly applicable to Mozi. However, Zhao Qi does not explicitly state this. In contrast to Zhao Qi, who only discusses the *two roots* concerning Yi Zhi, Zhu Xi believes that the *two roots* can refer to both Yi Zhi and Mozi. In other words, he explicitly broadens the scope of the *two roots*. As Zhu Xi (Li 1986, p. 1314) states, "*One root*, naturally, has many differences. *Two roots* simultaneously exist, and there is no difference. Mozi is also *two roots*. The question is: is this consistent with Mencius' original intention? Thus, it is imperative to understand the essence of *two roots*.

Zhu Xi also explained the passage from the *Mengzi* that mentions the term *two roots*. Zhu Xi (Zhu 1983, pp. 262–63) states:

Mencius said that the love for his brother's son was different from that of his neighbor's son. Everyone must be born from his parents and there is no difference, it is the principle of nature like the will of Heaven. Therefore, classified or unequal love is established, and extended to others. Now, as Yi Zhi said, he sees his parents as no more than passersby, but the order of bestowing love should start from here. What else could it be if not *two roots*? However, he knows what to choose between priorities. Nothing can extinguish the inherent clarity of the original mind of Yi Zhi. This is the reason why he can be aware of his mistake.

Zhu Xi's explanation highlights three points. The first pertains to the Confucian notion of graded love as emphasized by Mencius. Second, it is inevitable that there is only one source of all things; perhaps, Zhu Xi's interpretation in this regard is more profound than the commentary of Zhao Qi of the Han Dynasty. Third, Yi Zhi's remark "bestowing love begins with one's parents" differs from the original Mohist doctrine and could even be construed as contradictory.[12] In Zhu Xi's view, the fundamental basis of the Confucian notion of graded love is *one root*. Alluding to Yi Zhi's point of view, Zhu Xi (Li 1986, p. 1314) states: "what has difference, *one root* has difference, (it) is not forged". In other words, because of *one root*, there is graded love. However, we cannot say that because of graded love, there is *one root*. As Zhu Xi (Zhu 2001, p. 444) notes: "there are also those who take differentiated love as *one root*, although there is no big mistake, but the meaning is not complete. If it is said that graded love is because of *one root*, then it is possible. If it is said that *one root* is because of love with distinctions, then it is not possible". In other words, *one root* contains graded love, but graded love is not *one root*, yet it is an essential attribute of *one root*.

Common sense suggests that emotions are more intense among family or relatives than among other groups. Accordingly, the Confucian notion of graded love has a strong psychological foundation. Its opposite, undifferentiated love, is unnatural to human psychology. In Zhu Xi's interpretation, the Mohist views expressed by Yi Zhi are problematic. Zhu Xi (Zhu 2001, p. 444) states:

Now, Yi Zhi is talking about undifferentiated love, but it is not known where it originates, and he also sees his parents as different from the others. Distributing love in order is not contrary to righteousness. If we start bestowing love to our relatives, it is hard to know the origin of this love. What is the difference between *one and two roots*? Those who may say that bestowing love begins with the relatives are implicitly in line with *one root* of our Confucian texts. I think a tiny lapse can lead to a huge difference. People who hold this view also do not know what *one root* is.

Although Zhu Xi is sympathetic to Yi Zhi's assertion "bestowing love begins with one's parents", he also strictly defends the basic boundary between Mohism and Confucianism. Accordingly, Zhu Xi refutes the view that Yi Zhi's assertion "distributing love begins with one's parents" is implicit in Confucianism. In following Zhu Xi, it can be said that undifferentiated love is the basic attribute of the *two roots*. In short, the essence of *two roots* is undifferentiated love. From the point of view of moral value, Mozi refused to ascribe parents with higher status than strangers, but emphasized that an objective position of impartiality should be adopted between relatives and strangers. Like Immanuel Kant's (1997) view that "humanity is an end in itself", no one individual has a higher moral value than another, and everyone is equal in terms of moral value. For Confucians, there are thousands of strangers, but Yi Zhi sees his parents as no different from them. In this regard, Zhu Xi (Li 1986, pp. 1313–14) quips sarcastically: "undifferentiated love seems not only two roots, but perhaps, it is ten million roots". Zhao Qi and Zhu Xi do not agree with the undifferentiated love of Mozi and Yi Zhi. Moreover, Confucianism is not opposed to impersonal and objective moral positions; it also espouses the view of "treating all people equally". However, this view only applies to the public domain, or between strangers.

*3.3. Later Scholars' Understandings of Two Roots: The Conflict between Ethical Principles*

Like Zhu Xi, A.C. Graham also believes that *two roots* applies to both Yi Zhi and Mo Zi. However, his understanding of the essence of the *two roots* differs from Zhu Xi's. Rather than dwelling on the problem or difference between undifferentiated love and graded love, A.C. Graham argues that the *two roots* of Yi Zhi is about loving all people without distinction, and favoring one's own family. At the same time, he notes that Mencius regarded *two roots* as contradictory principles (see Graham 1989, p. 43). Arguably, the tradition of Western philosophy with its focus on logical analysis is in the back of Graham's mind. Accordingly, he points out that Yi Zhi cannot pursue two different ethical directions, i.e., *two roots*, simultaneously. Graham also thought that this was not only a problem with Yi Zhi, but the central problem of the Mohist school. As he states: "the Mohist [ . . . ] have the problem of reconciling an equal concern for everyone with greater care for parents and ruler than for others, the issue which led Mencius to accuse the Mohist Yi-tzu of having 'two roots'" (Graham 1989, p. 158). Indeed, Yi Zhi commits to two positions simultaneously: an impersonal and objective position that emphasizes impartiality and a personal view or individual position that rationally develops what is beneficial to the individual. However, the challenge is: how can these two positions be reconciled and balanced?

Like A.C. Graham, David Nivison also analyzed the *two roots* from the point of view of ethics. Nivison (1996) writes:

> While we must be cautious about what Mencius meant by 'one root' and 'two roots' (the commentators and translators have various suggestions) it seems entirely possible that he is talking about the basis of Yi Zhi's moral system, which he is criticizing as being double, insisting that, morally considered, a human as one of Heaven's creatures has just one 'root.' And that root for him has to be, of course, the 'heart' in its different aspects as dispositional 'hearts.' . . . Yi Zhi's trouble, then, would be that he has gotten into a mess by accepting guidance both from his 'heart' and from a set of doctrines that are unconnected with the 'heart.'

In brief, Nivison points out that Yi Zhi is torn by two forces: the natural emotions of the heart/mind or the love of family, and the doctrine of universal love. On the surface, Yi Zhi is in a divided state. Furthermore, Nivison differentiates between sensibility, which comes from the emotions arising in the heart/mind, and reason, which strives to transcend the bounds of sensibility to derive its own arguments.[13] Therefore, Nivison highlights the dilemma between rational arguments and the emotional heart/mind. Moreover, it is important to note that Nivison did not discuss whether Mozi's ethics also have *two roots*. However, his analysis indicates that *two roots* problem does not figure into Mozi's ethics.

After analyzing the representative viewpoint of the *two roots* problem, we deepen the understanding of the problem from the point of view of moral philosophy.

## 4. The Essence of Yi Zhi's *Two Roots*: The Dualism of Practical Reason

The debate between Mengzi and Yi Zhi highlights a crucial issue of moral philosophy, that is, the dualism/duality of practical reason. The discussion of this issue starts with Henry Sidgwick (1838–1900).

### 4.1. Sidgwick's Problem

Sidgwick was a famous utilitarian philosopher in 19th century England. Rawls also regarded him as a significant figure of classical utilitarianism. In his well-known book *The Methods of Ethics*, first published in 1874, Sidgwick attempts to integrate utilitarianism (universal hedonism), egoism (egoistic hedonism), and intuitionism into a systematic discourse. He discovers that he can integrate utilitarianism and intuitionism, but he cannot integrate utilitarianism and egoism. Moreover, it is between utilitarianism and egoism that the concept of "the dualism of practical reason" emerges. This concept is the dilemma of practical reason. In this book, Sidgwick mentions and explains the dualism of practical reason at least three times. In the *Preface to the Second Edition*, he proposes the concept above.

Besides, in a footnote in Book III: Chapter XIV, Sidgwick (1922, p. 405) also mentions it. However, it is in the *Preface to the Sixth Edition* that he presents a more systematic account. Sidgwick (1922, p. xviii) writes:

> I found he expressly admitted that 'interest, my own happiness, is a manifest obligation,' and that 'Reasonable Self-love' [is 'one of the two chief or superior principles in the nature of man']. That is, he recognized a 'Dualism of the Governing Faculty'—or as I prefer to say 'Dualism of the Practical Reason.'

In response to Sidgwick's dilemma and its moral philosophical implications, it has been commented that (Xu 2011, p. 19):

> Although Sidgwick tried to put forward a systematic theoretical defense for utilitarianism in *The Methods of Ethics*, he finally realized that utilitarianism could not avoid what he called 'the dualism of practical reason,' that is, the tension between the rational development of self-interest and the maximization of general welfare from an impartial point of view. His final judgment on utilitarianism constituted a starting point for later debates, forcing later philosophers to explore a series of issues related to the nature of morality, including the question of whether moral viewpoints must be strictly impartial.

At this point, it is necessary to describe practical reason. "Philosophically speaking, practical reason is our general capacity to reflect and decide how to act" (Xu 2011, p. 2). While the fundamental question of normative ethics is "what should I do?", practical reason provides justifications for one's actions. Thus, broadly speaking, morality becomes a part of practical reason. With this understanding of practical reason, it may be concluded that the dualism of practical reason to which Sidgwick refers is a confrontation between the maximizing consequences of utilitarianism through impartial calculation and the development of rational self-interest. It can also be summarized as a confrontation between utilitarianism and egoistic self-love (Chen and Guo 2008).

*4.2. The Possible Response of Yi Zhi in the Context of Modern Moral Philosophy*

The contemporary American philosopher Thomas Nagel extends Sidgwick's view by presenting the opposition, as well as the reconciliation of the personal and impersonal or social positions. In Nagel's (1991, pp. 3–4, 14, 21, 44, 52) view, the dualism between these two positions arises from the division, or the duality, of the self. This is a step further than Sidgwick. Moreover, following these two moral philosophers, it could be argued that while Yi Zhi develops the personal position from the impersonal position, Confucianism develops the impersonal position from the personal position. It is crucial to note that this more comprehensive account of Confucianism is found in Song Confucianism's theory of the unity or oneness of all things (万物一体 *wanwu yiti*; see Chen 2012).

From the perspective of moral philosophy, Yi Zhi's thoughts are in line with consequentialism. When examining actions in terms of their consequences, the action that leads to the greatest consequences must be followed. In contrast, Mencius' thoughts are in accordance with deontology since they emphasize obligations to loved ones. Generally speaking, Confucianism also accepts the principles of "generating the wellbeing of all people under heaven" and "eradicating the suffering of all people under heaven". However, maximization is not its goal (of course, it does not exclude maximization of benefits, whenever possible). For instance, Confucianism is critical of egoism or Yang Zhu's view, while rejecting the tendency of Mohist ethics to require maximizing the consequences of actions on other individuals. The latter point is similar to that of Bernard Williams (2006) who defends the individual position by alluding to the notion of personal integrity. In this regard, utilitarianism's principle of impartiality is in opposition to the rational development of human beings for their benefit. Thus, it undermines human integrity. Additionally, in Confucianism, the importance of individual points of view is relevant, but only when it is moral. Accordingly, Confucianism repudiates the Mohist principle of impartiality. Confucianism—particularly, Mencius—believes that maximization is not the goal of moral-

ity. Moreover, neither impartiality nor universal love is necessary in evaluating whether an action is morally right or wrong.

From the Confucian perspective, Sidgwick's method is problematic because it ignores effort or self-cultivation (功夫 *gongfu*). Moreover, in Western moral philosophy, the person is a rational being and believes there is only one right path among the many. The right path achieves people's unanimous consent. In contrast, Mencius emphasizes the unity of sensibility and reason (Wong 1991).[14] This unity ensures that a person can exert effort or cultivate the self because this is essential when acting in accordance with one's will. Moreover, for Confucianism, the demands of *ren ai* and universal love are the same. Although Mencius suggests that a person can start by loving one's family and then love others, there is no necessary or logical connection between graded love and comprehensive love (博爱 *bo ai*).[15] In fact, the former may also hinder the realization of the latter, and thus produce undesirable consequences such as nepotism or unfair treatment of others. In sum, from a theoretical perspective, Yi Zhi can cite Nagel's relevant thinking in response to Mencius' criticism. This proposal differs from the portrayal in the text where Yi Zhi succumbs to Mencius.[16]

### 4.3. Yi Zhi's Place in the History of Moral Philosophy

Yi Zhi faces the dilemma of practical reason because he tries to integrate the universal love (undifferentiated love) introduced by Mozi and the reasonable self-love of the individual (bestowing love begins with one's parents). Bentham, Mill and Mozi emphasize impartiality. Thus, there is no need to pay attention to an individual's reasonable self-love. In other words, the moral philosophy of Mozi and Mill has only one basic principle; therefore, it is *one root*. Perhaps, in their view, only Yi Zhi, a thinker who does not pursue theoretical thoroughness, could have an ethics that is *two roots*. Broadly, Confucian philosophy, as represented by Mencius, is deontological ethics.[17] Deontology emphasizes the fulfillment of moral obligations, but some obligations are not based on choice, but are determined by birth, such as obligations to family. Since these family obligations emerge because of the special relationship between family members, they must also be generalized or universalized. Accordingly, there is also only one basic principle of Mencius' moral philosophy—*one root*.

A criticism of deontology with respect to utilitarianism is that it is impersonal. Thus, it ignores the possibility of the subject or the person to act according to his relationship with others. However, considering the history of utilitarianism, there is a tendency to accommodate some considerations specific to the subject—the agent-relative. This is evident, for example, in the moral philosophy of David Sosa (1993). In the recent development of utilitarianism, the moral imperative of impartiality has been weakened by the belief that it is also moral to care for loved ones and friends, people with whom the subject has a special relationship. In this development, Sidgwick diverges from utilitarianism. In the same vein, Yi Zhi is no longer in line with classical utilitarianism, instead resembling the later development. Accordingly, Yi Zhi's modification of Mozi's ethics has a special place in the history of moral philosophy. In ancient China where the people accepted inequality, it was inevitable that Mozi's principle of universal love would be ignored. However, under modern conditions, the realization of Mohism's universal love is both possible and realistic. For example, Rawls (1999) argues that his principle of difference is an explanation of the principle of *fraternité*, which is part of the three principles of *liberté*, *égalité*, and *fraternité*. It is also believed by many that universal love is the Chinese version of fraternity or humaneness.

Thus far, there is no satisfactory answer to the question of whether Yi Zhi can get out of the dilemma of practical reason.[18] It may even be impossible to solve this problem in the field of moral philosophy. Perhaps, this problem can be addressed in political philosophy where the state will compensate for the losses of individuals whose actions are directed toward the benefit of the majority.

## 5. Conclusions

This article explores and commends the significance of Yi Zhi in moral philosophy. Yi Zhi is a Mohist and is regarded as the theoretical opponent of Mencius, a great Confucian scholar. In general, scholars do not distinguish the ethics of Mozi and Yi Zhi. This article argues that Mohist ethics have two versions: the original version of Mozi and the modified version of Yi Zhi. Mozi's emphasis on universal love (or impartiality) leads to neglecting the development of rational self-interest. Accordingly, Yi Zhi's remarks are a clarification or modification of Mozi's thoughts. First, Yi Zhi alluded to the concept of undifferentiated love to explain universal love as the basis of impartiality. Second, as he understood the concept of undifferentiated love in relation to the idea that "bestowing love begins with one's parents", Yi Zhi incorporated rational self-interest. Moreover, Mencius criticized Yi Zhi and disparaged his remarks as two roots (二本 er ben), contrasting it to Confucian ethics, which he said was one root (一本 yi ben). In fact, Yi Zhi and Henry Sidgwick, the founder of classical utilitarianism, face the same dilemma of practical reason: the conflict between utilitarianism and the self-interest of egoism. Mozi's ethics is agent-neutral, which means that it prioritizes the interests of the community. In contrast, Mencius' ethics is agent-relative, which means that it puts more importance on the integrity of the individual and opposes the unprincipled sacrifice of the individual for the sake of the community. Yizhi's ethics lies somewhere in between, emphasizing both the interests of the community and rational self-interest. Perhaps, looking at Yi Zhi's ethics helps explain the Confucian idea of the relationship between the individual and the community.

**Funding:** *A General History of Learning of Master Zhu in Ming and Qing Dynasties* (Major Project of The National Social Science Fund of China): 21&ZD051.

**Institutional Review Board Statement:** Not applicable.

**Informed Consent Statement:** Not applicable.

**Data Availability Statement:** Not applicable.

**Acknowledgments:** The assistance of my postdoctoral student, Mark Kevin S. Cabural, in translating this article is greatly appreciated.

**Conflicts of Interest:** The author declares no conflict of interest.

## Notes

[1]   In this article, we use the translations of D. C. Lau (2003) and Bryan W. Van Norden (2008). More particularly, in Mencius 3A5, we use Lau's translation; other passages from the *Mengzi* cited in this article are taken from Van Norden's translation.

[2]   Throughout most of this article, 爱无差等 is translated as "undifferentiated love" and 施由亲始 is translated as "bestowing love begins with one's parents".

[3]   Lao (2005) thinks that the primary principle of Mohism is utilitarianism.

[4]   There are, however, scholars who interpret Mohist ethics from the standpoint of divine-command theory (for instance, see Li 2006). I present three points to challenge or argue against such an interpretation or reading. First, "how can we know that God commands or forbids?" Mozi does not inform us. Second, "the Divine Command theory means that a conduct is right because and only because it is commanded by God". Indeed, in the *Mozi*, there are instances that promote egoism and utilitarianism, which are contrary to the divine-command theory that states the command of God is the only criterion of morality. Third, and most importantly, Mozi proposes that three criteria are the bases for judging right and wrong actions. God is not the origin of the three criteria, but humans are the rightful judge of actions. The *Mozi* (35.3) states: "You must establish standards [ . . . ] What are the three criteria? Master Mo Zi spoke, saying: There is the foundation; there is the source; there is the application. In what is the foundation? The foundation is in the actions of the ancient sage kings above. In what is the source? The source is in the truth of the evidence of the eyes and ears of the common people below. In what is the application? It emanates from government policy and is seen in the benefit to the ordinary people of the state. These are what are termed the 'three criteria'". (Note: For the first two points, please see Frankena 1973.)

[5]   We use Ian Johnston's (2010) translation.

[6]   Zhang Huiyan 张惠言 (1761–1820) of the Qing Dynasty was the first to propose that the essence of Mozi's teachings is universal love. This view was later endorsed by Sun Yirang 孙治让 and Liang Qichao.Moreover, one may argue that from the translation of "兼爱" as universal love has some Christian connotation. Since this article proceeds from a utilitarian reading of Mohist ethics, it

is imperative to explain why a term that seems close to divine-command theory has been adopted. In my defense, the usual translation of "兼爱" is universal love. For instance, Graham (1978) translates it with this very term and he likewise describes Mohist ethics as utilitarian.

7   For more information about the ten doctrines, see Loy (n.d.).

8   For instance, Mill (2015) states:"I have dwelt on this point, as being a necessary part of a perfectly just conception of Utility or Happiness, considered as the directive rule of human conduct. But it is by no means an indispensable condition to the acceptance of the utilitarian standard, for that standard is not the agent's own greatest happiness, but the greatest amount of happiness altogether".

9   We use Brook Ziporyn's (2009) translation.

10  We use the translation of John S. Major, Sarah A. Queen, Andrew Seth Meyer, and Harold D. Roth (Major et al. 2010), with additional contributions by Michael Puett and Judson Murray.

11  Dong (2015) also thinks that Yi Zhi modified the Mohist conception of universal love by referring to "love is without differentiation, but it is bestowed beginning with one's parents". However, Dong only mentioned it in passing, and he did not examine this modification from the point of view of moral philosophy.

12  Some modern interpreters have argued that Yi Zhi is inconsistent. As Lau (2003) notes, "by a dual basis, Mencius is presumably referring to the incompatibility between the denial of gradation of love and the insistence on its beginning with one's parents".

13  In fact, Yi Zhi's dilemma can also be explained through Nagel's point of view. In this regard, Yi Zhi is caught in the splitting of the self or the duality of positions, embodying the conflict/separation between personal and impersonal (or social) positions.

14  To avoid the Confucian idea of love from being understood as narrow, Wong (1991) rationalizes and generalizes emotions.

15  Li Jinglin argues that filial piety and love for kinship are the intermediaries between self-love and universal human love (see Li 2009).

16  Some commentators interpreted that Yi Zhi was persuaded and eventually became a Confucian based on two statements: first, at the end of the passage in the *Mengzi* wherein it says that "[Yi Zhi] who looked lost for quite a while and replied, 'I have taken this point'"; and second, from Zhu Xi's explanation that Yi Zhi became cognizant of his wrongdoing which motivated him to leave Mohism and embrace Confucian teachings (see Yang 2019). In this article, we show that Yang's argument is very limited.

17  Some researchers think that Confucian ethics is virtue ethics (see Huang 2020). According to Aristotle (2001), virtue forms based on habits, or is the result of repeated correct behavior. Right behavior refers to the question "how should I act?"—a question that is central to normative ethics (deontology and consequentialism). In this respect, virtue ethics cannot constitute an independent type of ethics. Accordingly, even if Mencius' ethics is regarded as virtue ethics, it also emphasizes that the right behavior is to bestow more love to relatives. For Mencius, this is not only the right behavior but also a virtue. In this way, Mencius' ethics is consistent with deontology in opposing Mozi's utilitarian ethics.

18  Nagel (1991, p. 5) also points out: "the problem of designing institutions that do justice to the equal importance of all persons, without making unacceptable demands on individuals, has not been solved—and that this is so partly because for our world the problem of the right relation between the personal and impersonal standpoints within each individual has not been solved". Although Nagel argues in the area of political philosophy, his argument is also applicable to moral philosophy. As Nagel, Nozick, and others have pointed out, political theory is partly an application of moral theory.

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
