# Peer review of "One or Two Roots? Yi Zhi and the Dilemma of Practical Reason"

_religions, doi:10.3390/rel13100885_

Round 1

Reviewer 1 Report

The paper revisits some early Chinese arguments about morality, and at times it almost suggests that it will bring some contemporary moral sensibilities into the discussion. Unfortunately, the paper never gets beyond a re-presentation of some components of the thought of Mencius that have been exhaustively written about over the course of the last decades, and the paper offers nothing original that could contribute to current debates. The author tends to throw in references to modern thinkers but they are quite general and without any substantial content, as if we should somehow already understand the author's train of thought, which we don't. I suggest that the author focus on some contemporary moral philosophers first, and then, using those findings, return to Mencius.

Reviewer 2 Report

While the originality of the thesis defended in this article should be appreciated, the topic is too ambitious for a short article. As a result, many controversial assumptions are left unexplained. I will indicate a few examples:

1. The author does not explain why we can reasonably assume that Mengzi's testimony about the two versions of Mohist ethics squares with the Mohism presented in the book Mozi.

2. The author quotes Mengzi, Zhuangzi, Huainanzi, Zhu Xi and so on, but barely refers to the book Mozi. He or she does not explain why they are more reliable than Mozi for understanding Mohism.

3. There are three dominant readings of Mohism - utilitarianism reading, consequentialism reading, and divine-command theory reading. The author sides with the first reading without giving explanations. 

4. It is indeed a common practice to adopt a specific reading without further explanations, as far as the reading is widely accepted among scholars. However, if the author wants to accept the utilitarianism reading, he or she must explain why he or she adopts translations that are nonetheless more in line with divine-command theory. For example, the author translates "jian ai" as "universal love"; the word "love," however, has certain Cristian connotations.

Having indicated the problems, however, I certainly do not want to downplay the originality of this study and the intellectual quality demonstrated by the article.

Round 2

Reviewer 1 Report

The paper is vastly improved and may be accepted in present form.

Reviewer 2 Report

While I may slightly disagree with the author on certain points, our disagreements are minor: they are to a large extent resulted from our different assumptions.

The revised version nicely addresses my concerns and incorporates my earlier comments.